



# Hydrographic fronts shape productivity, nitrogen fixation, and microbial community composition in the South Indian Ocean and the Southern Ocean

Cora Hörstmann[1,2], Eric J. Raes[3,1], Pier Luigi Buttigieg[4], Claire Lo Monaco[5], Uwe John[1,6], Anya M. Waite[7,1]

[1]Alfred Wegener Institute for Polar and Marine Science, Bremerhaven, Germany
[2]Department of Life Sciences and Chemistry, Jacobs University, Bremen, Germany
[3]CSIRO Oceans and Atmosphere, Hobart, Tasmania, Australia
[4]Helmholtz Metadata Collaboration, GEOMAR, Kiel, Germany
[5]LOCEAN-IPSL, Sorbonne Université, Paris, France
[6]Helmholtz Institute for Functional Marine Biodiversity, Oldenburg, Germany
[7]Ocean Frontier Institute and Department of Oceanography, Dalhousie University, Halifax, NS, Canada

*Correspondence to*: Cora Hörstmann cora.hoerstmann@awi.de

**Abstract.** Biogeochemical cycling of carbon (C) and nitrogen (N) in the ocean depends on both the composition and activity of underlying biological communities and on abiotic factors. The Southern Ocean is encircled by a series of strong currents and fronts, providing a barrier to microbial dispersion into adjacent oligotrophic gyres. Our study region straddles the boundary between the nutrient-rich Southern Ocean and the adjacent oligotrophic gyre of the South Indian Ocean, providing an ideal region to study changes in microbial productivity. Here, we measured the impact of C- and N- uptake on microbial community diversity, contextualized by hydrographic factors and local physico-chemical conditions across the Southern Ocean and South Indian Ocean. We observed that contrasting physico-chemical characteristics led to unique microbial diversity patterns, with significant correlations between microbial alpha diversity and primary productivity (PP). However, we detected no link between specific PP (PP normalized by chlorophyll *a* concentration) and microbial alpha and beta diversity. Prokaryotic alpha and beta diversity were correlated with biological $N_2$ fixation, itself a prokaryotic process, and we detected measurable $N_2$ fixation to 60° S. While regional water masses have distinct microbial genetic fingerprints in both the eukaryotic and prokaryotic fractions, PP and $N_2$ fixation vary more gradually and regionally. This suggests that microbial phylogenetic diversity is more strongly bounded by physical oceanographic features, while microbial activity responds more to chemical factors. We conclude that concomitant assessments of microbial diversity and activity is central in understanding the dynamics and complex responses of microorganisms to a changing ocean environment.



## 1 Introduction

The Southern Ocean (SO), and in particular its sub-Antarctic zone, is a major sink for atmospheric $CO_2$ (Constable et al. 2014). The SO is separated from the South Indian Ocean Gyre (ISSG) by the South Subtropical Convergence province (SSTC), comprising of the Subtropical Front (STF) and the Subantarctic Front (SAF). The SSTC is a zone of deep mixing and, thus, elevated nutrient concentrations (Longhurst, 2007). Further, the SSTC has been shown to act as a transition zone both numerically and taxonomically for dominant populations of marine bacterioplankton (Baltar et al., 2016).

In this dynamic context, a key driver of microbial productivity is nutrient availability, especially through tightly coupled carbon (C) and nitrogen (N) cycles. The constant availability of nutrients through vertical mixing in frontal zones, such as the STF, enhances primary productivity (Le Fèvre, 1987) and chlorophyll *a* (chl *a*) concentrations (Belkin and O'Reilly, 2009). Primary productivity (PP) and specific primary productivity ($P^B$ = primary productivity per unit chl *a*) are reflected in the relative abundance of different phytoplankton size classes whose productivity are, in turn, stimulated by nutrient injections via shallowing of mixed layer depth (MLD) at the SO fronts (Strass et al., 2002); decreasing the possibility of N-limitation. N-limitation can also biologically be alleviated through $N_2$ fixation mediated by diazotrophs; significantly contributing to the N pool in oligotrophic regions (Tang et al., 2019). In high-latitude regions, biological $N_2$ fixation could potentially have a large impact on productivity (Sipler et al., 2017). However, large disagreements exist between models of high-latitude $N_2$ fixation and its coupling to microbial diversity due to sparse sampling in these regions (Tang et al., 2019).

Due to the dynamics of the region, conflicting observations, and climate-driven changes, resolving the coupling of microbial productivity and diversity is particularly important across the strong environmental gradients crossing the ISSG, through the SSTC into the SO. Indeed, climate variability has been shown to impact ocean productivity, and thus influences the provision of resources to sustain ocean life (Behrenfeld et al., 2006). To date, observations of climate change-related effects in this region of the SO have been synthesized only based on long-term nutrient concentration and physical (temperature and salinity) changes (Lo Monaco et al., 2010), however, these typically lack a microbial dimension. Microbial composition, activity, and C export may all be impacted by climate-driven changes in ocean dynamics (Evans et al., 2011) such as MLD shallowing, eddy formation, and poleward shifts of ocean fronts (Chapman et al., 2020). For a more holistic, ecosystem-based understanding of this region, concomitant assessments of 1) steady-state biogeochemical processes through rate measurements of key elements (such as C and N) and 2) the microbial diversity that underpins it are essential enhancements to such long-term investigations.

Here, we measure the impact of C- and N- uptake on microbial community diversity, alongside the effects of hydrography (e.g., dispersal limitation) and local physico-chemical conditions across the Southern Ocean and South Indian Ocean. We focused our investigation on surface communities, aiming to resolve horizontal, surface variation. We used our observation to assess whether the following relationships - previously observed in related systems - hold in our study region:



*(1) Microbial diversity increases with increasing primary productivity (PP)*. Previous work has claimed that more resources support higher species richness, until intermediate rates of PP (Fig.1; Vallina et al., 2014) within ocean provinces (Raes et al., 2018).

(2) *Frontal systems are discrete ecological transition zones between regions*: To provide perspectives on the findings of Baltar et al. (2016; see above). These systems often separate water masses with distinct trophic structures (e.g. Albuquerque et al., 2021).

(3) Microbial alpha and beta diversity are impacted by $N_2$ fixation, itself correlated with the presence of other available sources of N and/or temperature: To provide more evidence on the role of $N_2$ fixation to the N budget in high latitudes (see eg. Shiozaki et al., 2018; Sipler et al., 2017).

To our knowledge, there are no concomitant evaluations of how surface gradients, microbial activity, and community composition relate to one another in this region. Here, we provide perspectives on these key relationships across the Southern Indian Ocean Subtropical Gyre (ISSG), the Subtropical Front (STF), and Subantarctic Front (SAF), and the SO comprising the Polar Front (PF) and Antarctic Zone (AZ).

## 2. Materials and methods

### 2.1 Study region, background data and sample collection

Our study region ranged from La Réunion Island in the Southern Indian Ocean Gyre (ISSG) to south of the Kerguelen Islands in the Southern Ocean (56.5° S, 63.0° E; Fig. 1a) as part of a larger repeated "OISO" sampling program – (Metzl 1998; https://doi.org/10.17600/17009700). Samples were collected as part of the VT153/OISO27 (MD206) cruise onboard the R/V *Marion Dufresne* from 2017-01-06 to 2017-02-07. Physical and biogeochemical data, as well as metadata, were collected from a rosette equipped with Niskin bottles and a Conductivity Temperature Depth (CTD) (Seabird SBE32) equipped with a SBE43 $O_2$ sensor and a Chelsea Aqua tracker fluorometer. OISO long-term data, starting in 1998, were used as a backdrop to our data collected in 2018 and allowed us to monitor changes in physical and chemical oceanographic properties over time (Supplementary A).

### 2.2 Province delineation after Longhurst

We identified three main clusters (i.e. ocean provinces) and five subclusters (i.e. water masses) on a temperature-salinity plot (Fig. 1b). As an overview, we used CTD depth profiles to validate the vertical extent of water masses in our samples (Fig. 1c,d) and checked the horizontal extent of the identified clusters using remote sensing data of sea surface temperature (Fig. S2). Additionally, we checked the horizontal boundaries of these clusters for matches in strong chl *a* concentration gradients



as an approximate for biological component of ocean provinces, following the concept of Longhurst (2007). Satellite data were
acquired from MODIS (https://neo.sci.gsfc.nasa.gov/), with images processed by NASA Earth Observations (NEO) in
collaboration with Gene Feldman and Norman Kuring, NASA OceanColor Group (Fig. S3). We calculated the geodesic
distance between sites from latitude/longitude coordinates using the geodist package in R (v0.0.4; Padgham et al., 2020).

## 2.2 Nutrient analysis

Dissolved inorganic nutrient concentrations, including phosphate ($PO_4^{3-}$), silicate (Si), mono-nitrogen oxides ($NO_x$), nitrite
($NO_2^-$), and ammonium ($NH_4^+$) were assayed on a QuAAtro39 Continuous Segmented Flow Analyser (Seal Analytical)
following widely used colorimetric methods (Armstrong, 1951; Murphy and Riley, 1962; Wood et al., 1967) with adaptations
to particular needs for Seal Analytical QuAAtro autoanalyzer. $NH_4^+$ was measured using the fluorometric method of Kérouel
and Aminot (1997). Detection limits of these methods were 0.1 µmol $L^{-1}$ for $PO_4^{3-}$ , 0.3 µmol $L^{-1}$ for Si, 0.03 µmol $L^{-1}$ for $NO_x$,
and 0.05 µmol $L^{-1}$ for $NH_4^+$.

## 2.3 Dissolved inorganic nitrogen and carbon assimilation

At each CTD station, water samples to measure primary productivity (PP) and $N_2$ fixation were taken from the underway flow-
through system (intake at 7 m). As the ship was moving during sampling, the distance between samples of the same station
can range up to ~15 km. Incubations were performed in acid-washed polycarbonate bottles on deck at ambient light conditions.
All polycarbonate incubation bottles were rinsed prior to sampling with 10% HCl (3x), deionized $H_2O$ (3x), and sampling
water (2x). In order to obtain the natural abundance of particulate nitrogen (PN) and particulate organic carbon (POC), which
we used as a t-zero value to calculate the assimilation rates, 4 L of water were filtered onto a 25 mm pre-combusted GF/F filter
for each station.
$N_2$ fixation experiments were carried out in triplicate for each station. We used the combination of the bubble approach
(Montoya et al., 1996) and the dissolution method (Mohr et al., 2010) proposed by Klawonn et al. (2015). 4.5 L bottles were
filled up headspace-free. All incubations were initialized by adding a $^{15}N_2$ gas bubble with a volume of 10 ml. We used $^{15}N_2$
labeled gas provided by Cambridge Isotope Laboratories (Tewksbury, MA). Bottles were gently rocked for 15 minutes. Finally,
the remaining bubble was removed to avoid further equilibration between gas and the aqueous phase. After 24 h, a water
subsample was transferred to a 12 ml exetainer® and preserved with 100 µL $HgCl_2$ solution for later determination of exact
$^{15}N$-$^{15}N$ concentration in solution. Natural $^{15}N_2$ was determined using Membrane Inlet Mass Spectrometry (MIMS; GAM200,
IPI) for each station with an average enrichment of 3.8 ± 0.007 atom % $^{15}N_2$ (mean ± SD; n=104) Primary productivity was
measured by adding $Na^{13}CO_3$ at a final $^{13}C$ concentration of 200 µmol $L^{-1}$.
Incubation bottles were incubated on board at ambient sea surface temperature (SST; water intake at 7 m) using a continuous
flow-through system. Temperature of both incubation bins was continuously measured. After 24 hours, the C and $N_2$ fixation
experiments were terminated by collecting the suspended particles from each bottle by gentle vacuum filtration through a 25



mm pre-combusted GF/F filter (<10 kPa). Filters were snap-frozen in liquid nitrogen and stored at -80° C while at sea. Filters
with enriched (T24) and unenriched (T0) samples were acidified and dried overnight at 60° C. Analysis of $^{15}$N and $^{13}$C
incorporated was carried out by the Isotopic Laboratory at the UC Davis, California campus, using an Elementar Vario EL
Cube or Micro Cube elemental analyzer (Elementar Analysensysteme GmbH, Hanau, Germany).
Carbon assimilation rates were calculated according to Knap et al. (1996), excluding the $^{14}$C - $^{12}$C conversion factor; and N$_2$
fixation was calculated according to Montoya et al. (1996), respectively. The minimum quantifiable rate was calculated
according to Gradoville et al. (2017).
**2.4 Pigment analysis**
For pigment analyses, 4 L of seawater were filtered (< 10 kPa) on 47 mm Whatman GF/F filter and stored at -80° C until
further analysis. High-Performance Liquid Chromatography (HPLC) was carried out as described in Kilias et al. (2013) with
following modifications: 150 μL of the internal standard canthaxanthin was included to each sample. Samples were dissolved
in 4 ml acetone and disrupted with glass beads using a Precellys 24 tissue homogenizer (Bertin Technologies, *France*) at 7200
rpm for 20 seconds. Detection of the sample at 440 nm absorbance using an HPLC analyzer (VARI AN Microsorb- MV 100-
3 C8). We used chl *a* concentration to estimate phytoplankton biomass. Pigment concentrations were calculated according to
Kilias et al. (2013), and quality controlled according to Aiken et al. (2009) (Supplementary A).
HPLC output data were analyzed using diagnostic pigments for the different taxa, and phytoplankton functional types (PFTs)
after Hirata et al. (2011) (Supplementary A, Table S2). This approach can be used to reveal dominant trends of the
phytoplankton community and size structure at the regional and seasonal scales (Ras et al., 2008). Furthermore, diagnostic
pigments were used to delineate three different size classes (pico-, nano-, and microplankton) according to Vidussi et al. (2001).
The relative proportion of each phytoplankton size class (PSC) was calculated based on the linear regression model proposed
by (Uitz et al. (2006). We investigated the patterns of PSCs with a second-order polynomial fit
(S1_code_archive/pigment_HPLC/diaganostic_pigments.R L143:153).
**2.5 DNA analysis**
Two liters of seawater from the shipboard underway system from each station were filtered through a 0.22 μm Sterivex® filter
cartridge for DNA isolation, snap-frozen in liquid nitrogen, and stored at -80 °C. DNA was extracted using a DNeasy® Plant
Mini Kit (QIAGEN, Valencia, CA, USA, Catalog No./ID: 69106) following the manufacturer's instructions. Sterivex
cartridges were gently cracked open and filters were removed and transferred into a new and sterile screw cap tube.
Approximately 0.3 g of pre-combusted glass beads (diameter 0.1 mm; 11079101 Bio Spec Products) and 400 μL Buffer AP1
were added to the filter, followed by a bead beating step using a Precellys 24 tissue homogenizer (Bertin Technologies, France)
with two times at 5500 rpm for 20 seconds with two minutes on ice in between and a final beat beating step at 5000 rpm for
15 seconds. DNA concentrations were quantified by the Quantus™ Fluorometer and normalized to 2 ng μL$^{-1}$.



### 2.5.1 Amplicon 16S and 18S rRNA gene PCR and sequencing

Amplicons of the bacterial 16S rRNA gene and eukaryotic 18S rRNA gene (using primers from 27F–519R; Parada et al. 2016, TA-Reuk454FWD1 – TAReukREV3; Stoeck et al. 2010, respectively) were generated following standard protocols of amplicon library preparation (16S Metagenomic Sequencing Library Preparation, Illumina, Part # 15044223 Rev. B; Supplementary B). 16S and 18S rRNA gene PCR products were sequenced using 250-bp paired-end sequencing with a MiSeq Sequencer (Illumina) at the European Molecular Biology Laboratory (EMBL) in Heidelberg (Germany) and at the Leibniz Institute on Aging (FLI) in Jena (Germany), respectively.

### 2.5.2 Amplicon sequence data analysis

For both 16S rRNA gene and 18S rRNA gene amplicon sequences, we used the DADA2 R package, v1.15.1 (Callahan et al., 2016) to construct Amplicon Sequence Variant (ASV) tables by following steps: Prefiltering filterandtrim function with truncL=50 and default parameters (S1_code_archive/dada2). Primer sequences were cut using the Cutadapt software implementation (v1.18) in the DADA2 pipeline, removing a fixed number of bases matching the 16S forward (515F-Y, 19 bp), reverse (926R, 20 bp), and the 18S forward (TA-Reuk454FWD1, 20 bp) and reverse (TAReukREV3, 21 bp) primers, respectively (S1_code_archive/dada2/dada2_16S.R L88:104; S1_code_archive/dada2/dada2_18S.R L92:104). Primer-trimmed fastq files were quality trimmed with a minimum sequence length of 50 bp, and checked by inspection of the average sequence length distribution (for both the 16S rRNA gene and 18S rRNA gene sequences). Samples within forward and reverse fastq files were dereplicated and merged with a minimum overlap of 20 bp. ASV tables were constructed and potential chimeras were identified de-novo and removed using the removeBimeraDenovo command. Sequencing statistics for removed reads and sequences in each step can be found in Table S3. Taxonomic assignment was performed using the SilvaNGS (v1.4; Quast et al. 2013) pipeline for 16S rRNA gene data with the similarity threshold set to 1. Reads were aligned using SINA v1.2.10 (Pruesse et al., 2012), and classified using BLASTn (v2.2.30; Camacho et al. 2009) with the Silva database (v132) as a reference database (Supplementary C). For taxonomic assignment of 18S rRNA gene amplicons, we used the Plugin 'feature-classifier' (from package 'q2-feature-classifier', v2019.7.0) in QIIME2 (Bokulich et al., 2018) and the pr2 database (v4.12; Guillou et al. 2013). We removed ASVs annotated to mitochondria and chloroplasts from 16S rRNA gene ASV tables, and ASVs annotated as metazoans from 18S rRNA gene ASV tables, respectively (S1_code_archive/import/import_16S.R L35:38; S1_code_archive/import/import_18S.R L29). ASV tables of 16S rRNA gene amplicon (Table S4) and 18S rRNA gene amplicons (Table S5) were used for further statistical analyses.

### 2.6 Ecological data and statistical analysis

A combination of temperature, salinity, dissolved oxygen concentrations, and dissolved inorganic nutrient concentrations ($NO_3^-$, $NO_2^-$, $NH_4^+$, Si, and $PO_4^{3-}$) were used to characterize the physical and biogeochemical environment of the study region. All statistical tests were performed in R version 3.6.3 (R Core Team, 2017). Statistical documentation, package citations and scripts are available in S1. Microbial alpha diversity was calculated with Hill numbers (Richness, Shannon entropy, Inverse



Simpson, q = 0 - 2; Chao et al., 2014) using the iNEXT package v2.0.20 in R with confidence set to 0.95 and bootstrap = 100
(S1_code_archive/alpha_diversity). Accordingly, rarefaction curves are shown in Fig. S6. Pearson correlations between
microbial richness (q=0), inverse Simpson diversity (q=2), environmental parameters, and biological rates were calculated and
plotted (ggplot2) (Fig. S7). P-values were adjusted for multiple testing using Holm adjustment (Holm, 1979), and residuals
checked for normal distribution (Fig. S8). For comparability and statistical downstream analyses, we performed the following
transformations to the ASV table and the environmental metadata: To account for the compositionality of sequencing data (see
Gloor et al. 2017), we performed a CLR-transformation for Redundancy Analysis (RDA). We used Hellinger transformation
(*decostand()* function in vegan) of the ASV pseudocount data (minimum pseudocount per ASV cutoff = 3) for PERMANOVA
analyses. Environmental data were z-scored for comparable metadata analysis (S1_code_archive/transformations). For
multivariate analyses of microbial beta-diversity and environmental parameters, we performed redundancy analyses (RDA) of
the CLR-transformed ASV tables (S1_code_archive/RDA). Differences of microbial beta-diversity (based on Hellinger
transformed ASV tables), phytoplankton community composition (based on pigment concentrations), and water masses were
tested with permutational ANOVA (PERMANOVA; Anderson, 2001) using the *adonis2()* function in vegan along with a beta
dispersion test to evaluate the homogeneity of dispersion (Fig. S9). To investigate at where differences of environmental
variables have an impact on microbial community dissimilarity, we performed a general dissimilarity model (GDM) of the
community dissimilarity and environmental variables, and checked for the influence of geographic distance based on spline
magnitude (gdm package; S1_code_archive/GDM).
As differences of microbial beta-diversity were significant in PERMANOVA analysis between provinces and water masses,
we performed a similarity percentage (SIMPER) analysis in R using the vegan package to assess which ASVs contribute most
to the observed variance of microbial community composition (Table S6; S1_code_archive/taxonomy_analyses). To determine
the number of ASVs shared between provinces (or unique to certain provinces), we transformed ASV pseudocount tables into
binary tables and calculated shared and unique ASVs using the upsetR package in R (v.4, Conway et al. 2017;
S1_code_archive/usetR). We calculated the percentage of all within-sample observed ASVs within the merged samples of a
province (Table S7).
**3. Results**
**3.1 Delimitation of regional water masses**
Through our analysis of temperature, salinity, oxygen and dissolved inorganic nutrient (N, P, Si) concentrations, we identified
five distinct water masses, fronts, and frontal zones: the ISSG, STF, SAF, PFZ, and AZ, which broadly aligned with three
oceanographic provinces (ISSG, SSTC and SO; Fig. 1a). Within the Southern Ocean (SO), we identified four water masses in
our transect including the Antarctic Zone (AZ) and three distinct frontal systems: 1) The Polar Front (PF), 2) The Subantarctic
Front (SAF), and 3) The Subtropical Front (STF; Fig. 1). In our analysis, stations 6, 7, and 9 were placed within the Polar





Front Zone (PFZ), between the SAF and PF. Due to the bathymetrically-driven convergence of the STF and SAF around
Kerguelen island, we consider the SAF as part of the convergence zone between the SO and IO, the South Subtropical
Convergence province (SSTC), rather than as a Southern Ocean frontal system. At 7 m depth, we noted clear shifts in
temperature (SST), salinity, and dissolved inorganic nutrient ($NO_3^-$, $PO_4^{3-}$, Si) concentrations when crossing the STF. The STF
is described as a circumpolar frontal zone creating the boundary between our measurements of the warm (20-25 °C), saline
(>35), and oligotrophic ($NO_3^-$ < 0.03 µM; $PO_4^{3-}$ : 0.04 - 0.21 µM) subtropical waters (STW) of the Indian South Subtropical
Gyre (ISSG) and the cold (3-6 °C), macro-nutrient rich ($NO_3^-$ : 19.2 - 24.9 µM; $PO_4^{3-}$ : 1.43 - 1.71 µM) SO (Fig. 1, Fig. 2, Fig.
S3). In the context of this study, STW and ISSG could be used interchangeably; we refer henceforth to ISSG.
**3.2 Primary productivity (PP)**
Maximum primary productivity (PP) within our dataset were measured near the Kerguelen plateau in the Polar Front Zone
(PFZ) at Station 9 (3236.8 and 3553.3 µmol $L^{-1}$ $d^{-1}$, respectively) and Station E (2212.4 - 2688.1 µmol $L^{-1}$ $d^{-1}$, n = 6). Comparing
all PP measurements across water masses, we found relatively high PP in other stations of the PFZ (Stations 6, 7; Fig. 3a;
Table 1) and in the Subantarctic Front (SAF) (Stations 4, 15). Lowest PP (190.4 - 642.6 µmol $L^{-1}$ $d^{-1}$) were measured at the
stations in the Indian South Subtropical Gyre (ISSG). While stations in the ISSG showed very little variations within one
station (e.g. 226.09 - 371.07 , n = 6, Station 18), variation within SO stations was relatively high (e.g. 587.42 - 1875.58 µmol
$L^{-1}$ $d^{-1}$, n = 6, Station 37; Table 1).
Overall, the variation of specific primary productivity ($P^B$) did not show great variations between provinces, with maximum
rates at station 11 (Table 1; Fig. 3b). We did not find a significant correlation between mixed layer depth and $P^B$ (Pearson
correlation; $r = 0.21$, $p = 0.47$, $n = 12$).
**3.3 $N_2$ fixation**
Di-nitrogen ($N_2$) fixation was above the minimum quantifiable rate (MQR) at all stations (Table 1). $N_2$ fixation measurements
did not show a clear temperature-dependent trend (Fig. 3), neither were they directly associated with low DIN values (Fig.
S10). $N_2$ fixation in the warm oligotrophic waters of the Indian South Subtropical Gyre (ISSG) was up to 7.93 nmol $L^{-1}$ $d^{-1}$
(Station 18; Fig. 3c; Table 1). Lowest $N_2$ fixations were measured in the productive zone of the STF and SAF (0.24 - 2.01
nmol $L^{-1}$ $d^{-1}$, n = 3). In the AZ, $N_2$ fixation ranged between 0.89 and 1.97 nmol $L^{-1}$ $d^{-1}$. The variation between replicates was
high, e.g. rates ranged between 0.9 to 7.9 nmol $L^{-1}$ $d^{-1}$ at station 18 (Table 1). Across provinces, we did not find notable
differences in $N_2$ fixation.
**3.4 Phytoplankton pigment analyses**
Photosynthetic pigment concentrations showed a clear separation between the oligotrophic ISSG and the nutrient-rich SO (Fig.
S5). Chlorophyll *a* concentrations were relatively low in the warmer water stations of the ISSG than in the SSTC and SO
(Table 1). The relative proportion of phytoplankton biomass to the total organic matter was estimated by calculating the ratio




of PN : chl *a* and showed a strong increase in the ISSG (11.5 - 29.7 PN : chl *a*, n = 4) in comparison to the SSTC (2.7 - 7.2 PN
: chl *a*, n = 3) and SO (2.8 - 15.3 PN : chl *a*, n = 6; Fig. S4).
The phytoplankton community composition was significantly and markedly different across provinces (PERMANOVA;
Permutations = 999, $R^2$ = 0.76, p < 0.001; n = 14) and water masses (PERMANOVA; Permutations = 999, p = 0.002; $R^2$ =
0.81, n = 14). The pigment concentration of prokaryote-specific pigment zeaxanthin was high in the ISSG (0.03 - 0.06 mg m$^{-3}$,
$^{3}$, n = 4; Fig. S5a). Zeaxanthin still occurred in the STF and SAF (0.03 - 0.04 mg m$^{-3}$, n = 3), but disappeared in the SO (< 0.01
mg m$^{-3}$, n = 6). *Prochlorococcus* was distinctly identified through its diagnostic pigment divinyl chl *a*, and showed a relatively
high pigment concentration in the ISSG (0.02 - 0.03 mg m$^{-3}$, n = 4; Fig. S5a). We found concentrations of diatom-specific
fucoxanthin (except station 18) ranging from 0.021 mg m$^{-3}$ in the ISSG (station 16) to 0.34 mg m$^{-3}$ in the SO (station 37; Fig.
S5a). Across water masses, fucoxanthin concentration was slightly higher in the AZ (0.06 - 0.5 mg m$^{-3}$, n = 4) than in all other
water masses (0 - 0.13 mg m$^{-3}$, n = 10).
The distribution of potential phytoplankton size classes (PSCs; pico- nano- and microplankton), calculated from diagnostic
pigments (Supplementary A), showed a clear pattern over temperature variations (Fig. S5b). The pigment data suggested that
picoplankton dominated warm water in the ISSG, picoplankton abundance sharply decreased (second-order polynomial fit $R^2$
= 0.98, p <0.001, n = 14) at lower values of SST. Pigment data also suggested that microplankton showed a contrary trend to
the relative fraction of picoplankton, having high abundance in cold-water and decreasing at higher values of SST, with a
minimum at 20°C SST and a slight increase (14% microplankton of all phytoplankton size classes) towards 25°C SST (second-
order polynomial fit $R^2$ = 0.77, p <0.001, n = 14). Nanoplankton showed a maximum at 12°C SST and decreased both towards
warmer and colder waters (second-order polynomial fit, $R^2$ = 0.58, p < 0.01, n = 14).
**3.5 Eukaryotic planktonic community composition**
For each station, except station 4, the V4 region of the small subunit ribosomal RNA gene (18S rRNA) was amplified and
sequenced to determine the community composition of micro-, nano-, and pico-eukaryotes in all three oceanic provinces. We
recovered a total of 2618 ASVs. After removing sequences annotated to metazoans, 2501 ASVs remained (4.4% of ASVs
removed).
We found a strong correlation between both eukaryotic richness and diversity (Inverse Simpson Index) with SST (Pearson
correlation, *r* = 0.85, p < 0.001 for Richness, and *r* = 0.82, p = 0.001 for Inv. Simpson, n = 12, respectively; Fig. S7a, S7c).
Overall, eukaryotic diversity was negatively correlated with PP (*r* = -0.66, p = 0.02, n = 12; Fig. S7e) and significantly and
positively associated with $N_2$ fixation (*r* = 0.74, p = 0.01, n = 12; Fig. S7g). However, a strong correlation between rate
measurements (PP, $N_2$ fixation) and eukaryotic diversity was only apparent in the ISSG, and no significant across other
provinces (Pearson correlation after removal of ISSG samples from dataset: PP *r* = 0.47, p = 0.24, and $N_2$ fixation, *r* = -0.48,
p = 0.23, n = 8, respectively).





Our RDA constrained 81% of the variance in the ASV table, with a p-value of 0.095 (Permutations = 999, n = 12). Sites were
well separated between Longhurst provinces along the first two RDA axes (capturing 52.67% constrained variance, Fig 4a).
Our PERMANOVA, which tested the province-based separation, produced moderate but significant results (Permutations =
999, $R^2$ = 0.54, p = 0.001; n = 12). An additional PERMANOVA grouping sites by water masses produced similar results
(Permutations = 999, $R^2$ = 0.67, p = 0.001, n = 12; Fig. 4a). We found that more ASVs only occurred in one province, rather
than in two or more provinces (Fig. 4e). Sites within the ISSG province were associated with SST and $N_2$ fixation. Sites in the
SSTC were associated with high $NH_4^+$ concentrations. Sites belonging to the SO were associated with dissolved inorganic
nutrients ($NO_3^-$, $PO_4^{3-}$, Si), dissolved oxygen, and chl $a$ concentrations as well as high PP. Linear relationships between beta
diversity and rates were only weak for PP (PERMANOVA; Permutations = 999, $R^2$ = 0.27, p = 0.004, n = 12) and both weak
and insignificant between beta diversity and $N_2$ fixation (PERMANOVA; Permutations = 999, $R^2$ = 0.13, p = 0.14, n = 12).
Investigating whether and at which magnitude environmental parameters have an effect on microbial community dissimilarity,
our general dissimilarity model (GDM) showed the expected curvilinear relationship between the predicted ecological distance
and community dissimilarity (Fig. 4c I). Based on I-spline magnitudes of all tested environmental variables, geographic
distance had little effect on community dissimilarity (Fig. S11a). Community dissimilarity changed most notably in response
to variability in low magnitudes of PP (i.e. ISSG and STF; 17% of total community dissimilarity, n =12) and plateaued with
PP above 1100 µmol C $L^{-1}$ $d^{-1}$ (Fig. 4c III). A community dissimilarity change occurred most notably when $N_2$ fixation when
rates were above 2 nmol $L^{-1}$ $d^{-1}$ (~ 19% of change in total community dissimilarity associated to changes in $N_2$ fixation rates.
Fig. 4c IV). Among all tested environmental parameters, our I-spline results showed that community dissimilarity increased
most in response to variability in MLD and $PO_4^{3-}$ concentrations (49% of change in total community dissimilarity associated
to MLD variability, and 63% to $PO_4^{3-}$ variability, respectively, n = 12; Fig. S11a).
Significant differences in community dissimilarity structure between Longhurst provinces were associated with high-
pseudocount taxa, dominated by dinoflagellates (Dinophyceae) and diatoms (Bacillariophyta; SIMPER analysis; Table S6).
The pseudocount of ASVs belonging to the phylum Ochrophyta (Bacillariophyta_X) contributed to differences between ocean
provinces (contributing to at least 9.51% of the differences in community dissimilarity between the SO and ISSG). Moreover,
4.79% of the differences in community dissimilarity between the SO and the SSTC were associated with a higher ASV count
of Bacillariophyta_X ASVs in the SO. Further, we identified ten ASVs belonging to the phylum Dinophyceae contributing
with 2.1% to the community dissimilarity structure between the SO and ISSG; and with 5.79% to the community dissimilarity
structure between the SSTC and ISSG. This was further supported by relatively high concentrations of the photosynthetic
pigments chl c3 and peridinin (both indicative pigments for dinoflagellates) in the SO and SAF. We found a relatively high
number of ASV94 and ASV23 (*Chloroparvula pacifica*) in the SSTC, contributing 3.07% to the community dissimilarity
between the SSTC and the ISSG.





### 3.6 Prokaryotic community composition

From each of 14 stations, a fragment of the small subunit ribosomal RNA gene (16S rRNA) was amplified and sequenced to obtain insights into the diversity and community composition of prokaryotes. A total of 1308 ASVs was recovered from which we removed 267 ASVs annotated as chloroplasts and 68 ASVs annotated as mitochondria. Prokaryotic richness increased with increasing sea surface temperature (Pearson correlation: $r = 0.65$, p-value = 0.03, n = 11; Fig. S7a). Maximum alpha diversity (Inverse Simpson) estimate was found in the SAF (81.92, Station 15; Fig. S7d). Prokaryotic alpha diversity (Inverse Simpson) was positively (but not significantly) linked to primary productivity ($r = 0.36$, p = 0.55, n = 11; Fig. S7f) but showed a significant negative correlation with $N_2$ fixation ($r = -0.7$, p = 0.05, n = 11; Fig. S7h).

Our RDA of the prokaryotic ASV table captured 90% of the total variance with a p-value of 0.06 (Permutations = 999, n = 11). Sites clustered into Longhurst provinces along the first two RDA axes (62.48% of variance constrained; Fig 4b). This was also shown in the PERMANOVA solution for Longhurst provinces (Permutations = 999, $R^2 = 0.62$, p < 0.001, n = 11) and our PERMANOVA grouping into water masses (Permutations = 999, $R^2 = 0.74$ p < 0.001, n = 11; Fig. 4b). We found more ASVs occurring in either the ISSG or the SO provinces rather than across all provinces (Fig. 4f). Further, the ISSG and the SO shared the least ASVs (Fig. 4f). In the RDA, sites within the ISSG province were positively associated with SST and $N_2$ fixation. Sites belonging to the SO were positively associated with dissolved inorganic nutrients ($NO_3^-$, $PO_4^{3-}$, Si), dissolved oxygen, and chl $a$ concentrations as well as high PP (Fig. 4b). The community composition within the SSTC (STF and SAF) was distinct from that of the ISSG and SO along the 2$^{nd}$ RDA axis (21.67% variance constrained) and positively associated with $NH_4^+$ concentrations (Fig. 4b). Linear relationships between beta diversity and rates were weak for PP (PERMANOVA; Permutations = 999, $R^2 = 0.31$, p = 0.007, n = 11) and $N_2$ fixation (PERMANOVA; Permutations = 999, $R^2 = 0.2$, p = 0.05, n =11).

Investigating whether and at which magnitude environmental parameters have an effect on prokaryotic microbial community dissimilarity, our general dissimilarity model (GDM) showed the expected curvilinear relationship (Fig. 4d I). Based on I-spline magnitude, geographic distance had little effect on community dissimilarity. The largest magnitude in community dissimilarity could be observed between 190 - 1200 µmol C L$^{-1}$ d$^{-1}$ (Fig. 4d III). Community dissimilarity changed most notably in response to variability in low magnitudes of $N_2$ fixation and did not change in samples with highest average $N_2$ fixation measurements (2.8 nmol L$^{-1}$ d$^{-1}$ Station 3, and 4.0 nmol L$^{-1}$ d$^{-1}$ Station 18, respectively). Largest magnitudes of community dissimilarity were associated with dissolved oxygen concentrations (Fig. S11b).

Taxonomically, based on analysis of the CLR-transformed ASV table, the prokaryotic community was dominated by Proteobacteria, Cyanobacteria, and Bacteroidetes, which are all typical clades for surface water samples (e.g. Biers et al., 2009). The greatest community differences occurred between stations of the Southern Ocean (SO) and the Indian South Subtropical Gyre (ISSG) provinces. Structure in community dissimilarity between the ISSG and SO were mostly associated with the number of Flavobacteriaceae (11.52% of total community dissimilarity, SIMPER analysis, Table S6) and





Alphaproteobacteria (5.69% of the total difference in community dissimilarity, SIMPER analysis, Table S6). The ISSG was
characterized by a high number of Cyanobacteria and some Actinobacteria. The cyanobacterial fraction was dominated by
*Prochlorococcus* and *Synechococcus,* respectively.
Within the class level, all stations were dominated by Alpha-, and Gammaproteobacteria, Bacteroidia, Oxyphotobacteria
(Cyanobacteria), and Verrucomicrobia. Within the Alphaproteobacteria, we found a great dominance of ecotype I, II, and IV
of SAR11 clade throughout all samples (Table S4). The relative number of pseudocounts of bacteria belonging to the phylum
Bacteroidetes decreased towards warmer SST in the ISSG, with significant differences between the SO and ISSG (Welch two
sample t-test t = 4.58, p < 0.001, n1 = 341, n2 = 151). The phylum Bacteroidetes was largely dominated by the order
Flavobacteriales (90.98% of annotated ASVs). Cyanobacteria mainly occurred in the SSTC and in the ISSG, which were
dominated by *Prochlorococcus* in the ISSG and *Synechococcus* in the SSTC, respectively. Cyanobacterial pseudocounts were
significantly lower in the SO in comparison to the SSTC (Welch two sample t-test, t = -3.86, p-value < 0.001, n1 = 17, n2 =
31) and to the ISSG (Welch two sample t-test, t = -4.74, p < 0.001, n1 = 17, n2 = 45). *Atelocyanobacteria* (UCYN-A) ASVs
occurred in the SAF (Station 14) and ISSG (Station 2, 3).
**4. Discussion**
Each water mass in our study had a distinct microbial fingerprint, including unique communities in frontal regions. We
highlight clear relationships between microbial diversity, and primary productivity and $N_2$ fixation (high linear and nonlinear
covariability) in the South Indian Ocean Gyre (ISSG), the Southern Ocean (SO), and their frontal transition zone.  Below, we
discuss how this clear provincialism of microbial diversity is dis-connected from regional gradients in primary productivity
(PP) and $N_2$ fixation across our transect. This could suggest that microbial phylogenetic diversity is more strongly bounded by
physical oceanographic boundaries, while microbial activity (and thus, perhaps, their functional diversity, not assessed here)
responds more to chemical properties that changed more gradually between the low- and high-nutrient provinces we sampled.
**4.1 $N_2$ fixation and associated microbial diversity display distinct regional variations**
Overall, our $N_2$ fixation (up to 4.4±2.5 nmol L$^{-1}$ d$^{-1}$) was comparable to $N_2$ fixation measured by González et al. (2014) above
the Kerguelen Plateau (up to 10.27±7.5 nmol L$^{-1}$ d$^{-1}$) and showed a similar latitudinal trend as $N_2$ fixation further east in the
Indian Ocean, however, around 10-fold lower absolute rates (0.8 - 7 vs 34 - 113 nmol L$^{-1}$ d$^{-1}$; Raes et al. 2014). We note that
the localized rates reported by González et al. (2014) are to date the only published $N_2$ fixation measurements in this region,
likely to be close to the annual maxima because of high irradiance, however, further investigations across seasonal changes
within the study area are needed to confirm our observations. Our regional data are therefore important in closing the gaps in
$N_2$ fixation measurements in the Southern Ocean, especially considering that large disagreements exist between models of
high-latitude $N_2$ fixation rates (Tang et al., 2019).



N₂ fixation measurements often show high basin-wide variability as well as high variability between samples at the same site,
being sensitive to details of experimental design, incubation, and sea-state conditions (Mohr et al., 2010). In aggregate, these
issues are best accounted for by calculating the minimum quantifiable rate (MQR; Gradoville et al., 2017). We observed high
heterogeneity of biological samples taken from the underway flow-through-system 5 minutes apart (separated by ~15 km)
within the same water mass. Similar variability in absolute measurements of N₂ fixation (2.6 - 10.3 nmol L$^{-1}$ d$^{-1}$ ± 7.5 nmol L$^{-1}$ d$^{-1}$)
were reported by González et al. (2014) close to our sampling site around Kerguelen Island. This could imply a
submesoscale variability or influence of other unmeasured parameters.
As oligotrophic gyres extend and displace southwards under climate change, (Yang et al., 2020), the biogeochemical and
physical characteristics of the SO are changing (Caldeira and Wickett, 2005; Swart et al., 2018), biological regional N₂ fixation
might become an important N-source for productivity. Our data showed maximal N₂ fixation in the oligotrophic waters of the
ISSG, however, notably, measurable N₂ fixation occurred well into the SO, to 56° S, suggesting that N₂ fixation contributes to
the regional N pool, despite other available sources of N (Shiozaki et al., 2018; Sipler et al., 2017). Similarly, we found a
negative N* in the SO, which potentially indicates a P excess supporting N₂ fixation (Knapp, 2012). Noteworthy is a slight
increase in N₂ fixation in the Antarctic Zone (AZ). High-latitude measurements in northern polar regions (Bering Sea) reached
10 -11 nmol N L$^{-1}$ d$^{-1}$ (Shiozaki et al., 2017), substantially higher than our measurements of the SO (0.8 - 1.9 nmol N L$^{-1}$ d$^{-1}$),
potentially supported by the close proximity to the coast or other factors such as day length, seasonality, diazotroph community
or trace metal concentrations.
Our results suggest that regional N₂ fixation was not limited by the presence of other sources of bioavailable N (Fig. S10), a
conclusion also reached in a number of studies including culture experiments (Boatman et al., 2018; Eichner et al., 2014;
Knapp, 2012), as well as in situ measurements in the South Pacific (Halm et al., 2012), off the coast of Chile and Peru with
rates up to 190 µmol m$^{-2}$ d$^{-2}$ (Fernandez et al., 2011), and across the Eastern Indian Ocean (Raes et al., 2015). This evidence
counters the hypothesis of Breitbarth et al. (2007) that N₂ fixation occurs only when other sources of N are limited. The
contribution of N₂ fixation to the N-pool – and thus to productivity – varies strongly with ecosystem structure: In the SO,
despite the local N₂-fixation measurements, N₂ fixation remains likely a very minor contributor to the N required by the
microbial community for primary productivity.
Our results also strongly suggest that prokaryotic community structure and composition (beta diversity) were strongly impacted
by the presence of biological N₂ fixation, itself a prokaryotic process (Karl et al., 2002). For example, the N₂-fixing
*Atelocyanobacteria* (UCYN-A) occurred in the SAF and ISSG; however, to gain a clear insight into the community and N₂
fixation, the diazotrophic community would need to be further resolved by amplicon analysis of functional (*nifH*) genes (Luo
et al., 2012) as shown in other high-latitude studies (Fernández-Méndez et al., 2016; Raes et al., 2020).



**4.2 Total and specific primary productivity differentially affect microbial diversity**

We found PP was highest in the PFZ and decreased towards higher latitudes in the SO (Fig. 3a). Strass et al. (2002) showed that frontal maxima of PP are expected, and the observed decrease was probably due to Fe limitation in the SO (Blain et al., 2008). Primary productivity can also be limited by Si concentration and light availability when the mixed layer deepens (Boyd et al., 2000), but in our data Si concentrations were high in the surface water samples, and light levels were close to maximum in austral summer. The measured maximum PP above the Kerguelen Plateau (station E) was likely stimulated by Fe inputs (Blain et al., 2007).

Our results did not support prior observations that frontal regions (SAF and STF) supported higher specific primary productivity ($P^B$) (as reported in the Antarctic Atlantic sector; Laubscher et al. 1993): While phytoplankton community composition, phytoplankton size distribution, and nutrient concentrations were strikingly different between the ISSG and SO, we found little difference in $P^B$, with some slightly lower values observed within the SSTC (Fig. 3b). Differences in $P^B$ usually arise from physiological changes due to variabilities in irradiance (Geider, 1987), nutrient concentrations (Behrenfeld et al., 2008; Chalup and Laws, 1990), or differences in phytoplankton community structure, where cyanobacteria have the highest PP efficiency and diatoms the lowest (Talaber et al., 2018). Thus, our observations suggest that either 1) there is a lack of selective pressure on photosynthetic efficiency between provinces or 2) mechanisms driving $P^B$ are different between provinces, and the sum of beneficial (e.g. increased nutrient concentrations in the SO) and detrimental mechanisms (e.g. low irradiance and photoinhibition through deep vertical mixing, reported from the ACC, Alderkamp et al., 2011) result in similar $P^B$. The slight variation around the frontal system is hard to interpret, as the complex interplay between factors may result in stochasticity.

Primary productivity can be an important driver for (phylogenetic) microbial alpha diversity (Vallina et al., 2014) especially within ocean provinces (Raes et al., 2018). While our observational study only has a small number of samples within and between oceanic provinces (n = 12, $n_{ISSG}$= 4, $n_{SSTC}$ = 3, $n_{SO}$ = 4), it did suggest that further validation of this assumption is needed. We observed that PP changed gradually across the sampling region, and that local variability in PP was high between samples taken ~15 km apart within the SSTC and SO (Fig. 3a). These local variabilities can arise from complex physico-chemical interactions between the STF, SAF and SO (Mongin et al., 2008). Counter to Vallina et al. (2014) and Raes et al. (2018), we found a negative correlation between eukaryotic alpha diversity and PP within the ISSG. Further, we found no correlation between eukaryotic diversity and PP within the SSTC and SO and none between prokaryotic alpha diversity across all provinces (Fig. S7).

In terms of beta diversity, we observed a structuring effect of PP for both pigment-, 16S rRNA gene-, and 18S rRNA gene-derived diversity profiles (Fig. 4 a,b, Fig. S5). Pigment analysis revealed that photosynthetic prokaryotic diversity is strongly impacted by the relative abundance of *Prochlorococcus,* which does not generally occur in cold, high-latitude waters (>40°S/N; Fig. S5) (Partensky et al., 1999) and, if so, only in low abundance (reviewed in Wilkins et al. 2013). Our 16S rRNA gene





analyses confirm these observations showing that 1) picoplankton - and specifically *Prochlorococcus* - had relatively high
proportions in the ISSG but very low in the SSTC, 2) *Synechococcus* dominated the Cyanobacterial fraction in the SSTC, and
3) both *Prochlorococcus* and *Synechococcus*, were not detected in the SO (Table S4, Table S6). In the SSTC and SO,
phytoplankton communities had high proportions of dinoflagellates (Dinophyceae) and diatoms (Bacillariophyta) (up to 74%
of diatom diagnostic pigment concentrations), which are known as essential contributors to marine PP and microbial diversity
(Malviya et al., 2016) and known to dominate the phytoplankton fraction within the Polar Frontal Zone (PFZ), especially as
the blooming season progresses (Brown and Landry, 2001).
Further, our results show that phytoplankton community structure appears to be tightly coupled to the occurrence of specific
heterotrophic organisms (Table S6), and thus may mediate an indirect effect of PP through microbial food webs (as also noted
in, e.g., Sarmento and Gasol 2012). For example, in areas of relatively high diatom concentrations, we found increased
proportions of Flavobacteria. These bacteria specialize on successive decomposition of algal-derived organic matter (Teeling
et al., 2012) and are known associates of diatoms (Pinhassi et al., 2004).

### 4.3 Implications for microbial regionality

Microbial diversity was regionally constrained independent of geographical distance (GDM analysis), but was partitioned into
ocean provinces as repeatedly described for other ocean basins such as the Pacific (Raes et al., 2018) and the Atlantic Ocean
(Milici et al., 2016). This supports the classical concept of microbial biogeography (Martiny et al., 2006). Further, we found
that microbial beta diversity was even better resolved by individual water masses, highlighting the importance of including
oceanographic boundaries that limit cross-front dispersal (Hanson et al., 2012; Hernando-Morales et al., 2017; Wilkins et al.,
2013a).
Our beta diversity analysis confirmed the findings by Baltar and Arístegui (2017) who found unique environmental sorting
and/or selection of microbial populations in the SAF and STF. Further, we were able to link these communities to high $NH_4$
concentrations. This suggests high recycling of nitrogen sources within the microbial loop, and potentially favoring
nitrification in this area (Sambrotto and Mace, 2000). We also found increased Dinoflagellate concentrations (PFT) which
have been described to grow well under $NH_4$ conditions (Townsend and Pettigrew, 1997). Despite our small sample size within
the SAF and STF, we were able to detect these characteristics, supporting the call from Baltar et al. (2016) of better integrating
frontal zones in our understanding of microbial biogeography.
Different trade-offs such as nutrient limitation and grazing can shape the microbial seascape (Acevedo-Trejos et al., 2018). In
our study, the deviation between PN : chl *a* was large between the So and IO with high PN : chl *a* ratios in the ISSG (Fig. S4),
which has been used as an indicator of a relatively high abundance of heterotrophic microbes and protists over autotrophic
organisms (Crawford et al., 2015; Hager et al., 1984; Waite et al., 2007). This would suggest that grazers formed a higher





fraction of total biomass in the ISSG than in the SO. However, we did not measure zooplankton biomass or grazing rates, so
this remains speculative.
**5. Conclusion & outlook**
Our study leads us to conclude that simultaneous assessment of microbial diversity, biogeochemical rates, and the physical
partitioning of the ocean (provincialism) is central to the understanding of microbial oceanography.
Each water mass in our study had a distinct microbial fingerprint, including unique communities in frontal regions. Microbial
alpha diversity and community dissimilarity correlated with biogeochemical rate measurements; however, mechanisms driving
this association need further investigation through high-resolution sampling across spatial and temporal scales. Our results also
indicate that high-latitude $N_2$ fixation could meaningfully contribute to the global and regional N-pool (as reported for Arctic
$N_2$ fixation by Sipler et al., 2017), which may become especially significant as global stratification (and concomitant
restrictions in deep water replenishment of nutrients) intensifies.
While our sampling is too limited to conclude the point, our observations that phylogenetic diversity is constrained by
hydrographic properties and province boundaries, but biogeochemical rates and nutrient concentrations are changing more
gradually suggests that trans-province functional redundancy is present despite strong biogeographic separation in
phylogenetic terms. As an outlook, we therefore encourage examining both phylogenetic and functional diversity to assess
how functional groups and guilds contribute to the major biogeochemical (C, N) cycles across provinces and other
biogeographic regions. Coordinated studies across ocean provinces are key to establishing the baselines we need to monitor
the rapidly changing properties of the Southern high-latitudes in the face of rising temperature, acidification, and perturbations
in regional currents.
**Code availability**
All  code  is  available  under  S1_code_archive.zip  and  additionally  publically  archived  under
https://github.com/CoraHoerstmann/MD206_Microbes/releases/tag/v1.0
**Data availability**
All HPLC data, environmental and rate measurement data, including PN, MIMS data, PP, $N_2$ fixation, and minimum
quantification rate calculations are stored at the PANGAEA database (Hörstmann et al. 2018). All sequences are archived in
the European Nucleotide Archive (primary accession: PRJEB29488).





**Author contribution**

CH did the post-voyage processing and analysis of all samples and wrote the manuscript. ER conducted the fieldwork, designed the experiments and contributed to data analysis and writing the manuscript. PLB contributed to data analysis, ecological interpretation, and writing the manuscript. CLM provided the historic physical and chemical data and contributed to the write-up. UJ helped with the DNA sequencing and writing the manuscript. AW contributed to design of the experiments, data analysis, and writing the manuscript.

**Conflict of interest statement**

No conflict of interest.

**Acknowledgment**

We thank Nicolas Metzel as well as the captain and crew of the Marion Dufresne. We thank Dr. Gaute Lavik from the Max Planck Institute of Marine Microbiology in Bremen for the guidance and allowance of using the membrane inlet mass spectrometry. We thank Stefan Neuhaus for his knowledge about the bioinformatics pipeline. We thank Dr.Vladimir Benes and his team from the Genomics Core Facility, European Molecular Biology Laboratory, Heidelberg, Germany, for their kind guidance and support with the 16 rRNA gene sequencing. We thank the Leibniz Institute on Aging (FLI) in Jena (Germany) for their support in 18S rRNA gene sequencing. We thank Dr. Allison Fong and Prof. Dr. Matthias Ullrich for their comments on this study.

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



**Figures**

**Figure 1: (a) The MD206 transect and OISO stations. Stations are colored according to water masses and encircled by sampling**
**extent: black circles indicate stations where only CTD (conductivity, temperature, depth) data is provided, and stations encircled in**



red denote where additional samples for C, N and community composition were taken. (b) A plot of potential temperature (in degrees Centigrade (°C) and salinity (in practical salinity units) using sea surface (7 m) data of the stations used in further microbial and C/N analyses. The yellow circle highlights the Indian Ocean gyre (ISSG), light blue circle the Subtropical Front (STF), blue circle the Subantarctic Front (SAF), dark green circle the Polar Front Zone (PFZ) and the light green circle indicates the Antarctic Zone (AZ), dashed lines indicate water masses clustered within ocean provinces: blue line marks the Subtropical Convergence province (SSTC) and green line the Southern Ocean (SO); (c) and (d) show depth profiles of temperature, oxygen and salinity along two transects of the OISO stations. Colored bars indicate water masses according to (b). (c) shows the western transect covering OISO stations 2, 3, 4, 5, 6 and 37 around 53±1°E longitude; (d) shows the eastern transect of OISO stations 10,11,12,13,14,15,16 and E around 68±5° E.

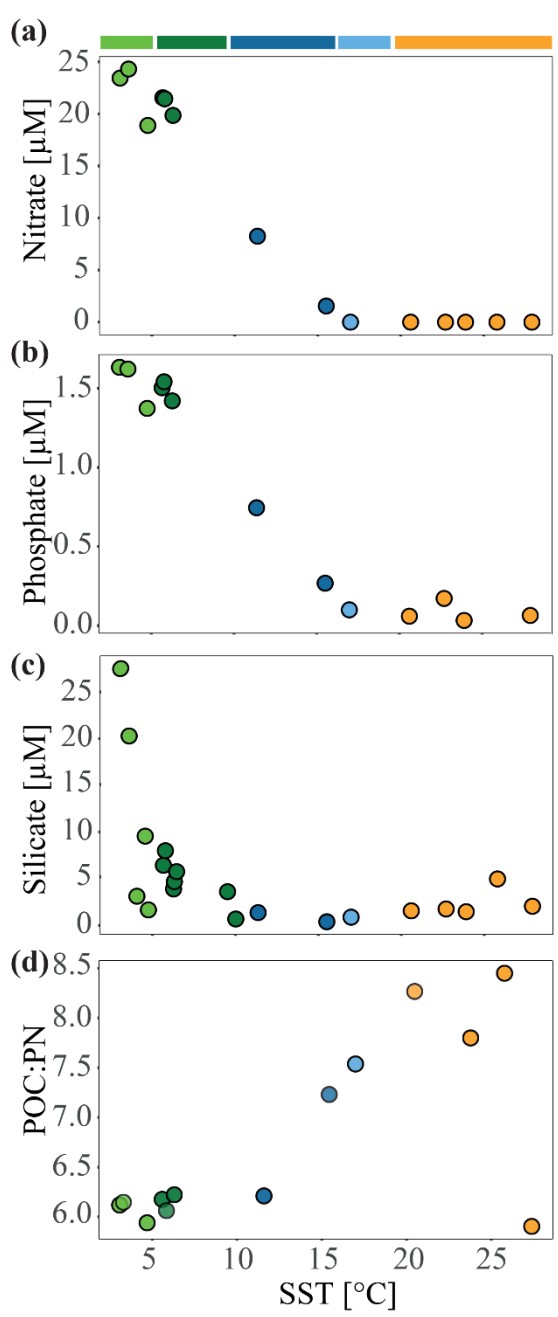

772

**Figure 2: Nutrient concentrations (µmol L⁻¹) and molar ratios of particulate organic carbon (POC) to particulate nitrogen (PN)**

**during the MD206 expedition against sea surface temperature (°C): (a) nitrate, (b) phosphate, (c) silicate, (d) POC:PN ratio. Colored**

**bars indicate water masses according to their sea surface temperature; yellow bar highlight the Indian Ocean gyre (ISSG), light blue**

**bar the Subtropical Front (STF), blue bar the Subantarctic Front (SAF), light green bar the Polar Front Zone (PFZ) and dark green**

**the Antarctic Zone (AZ).**



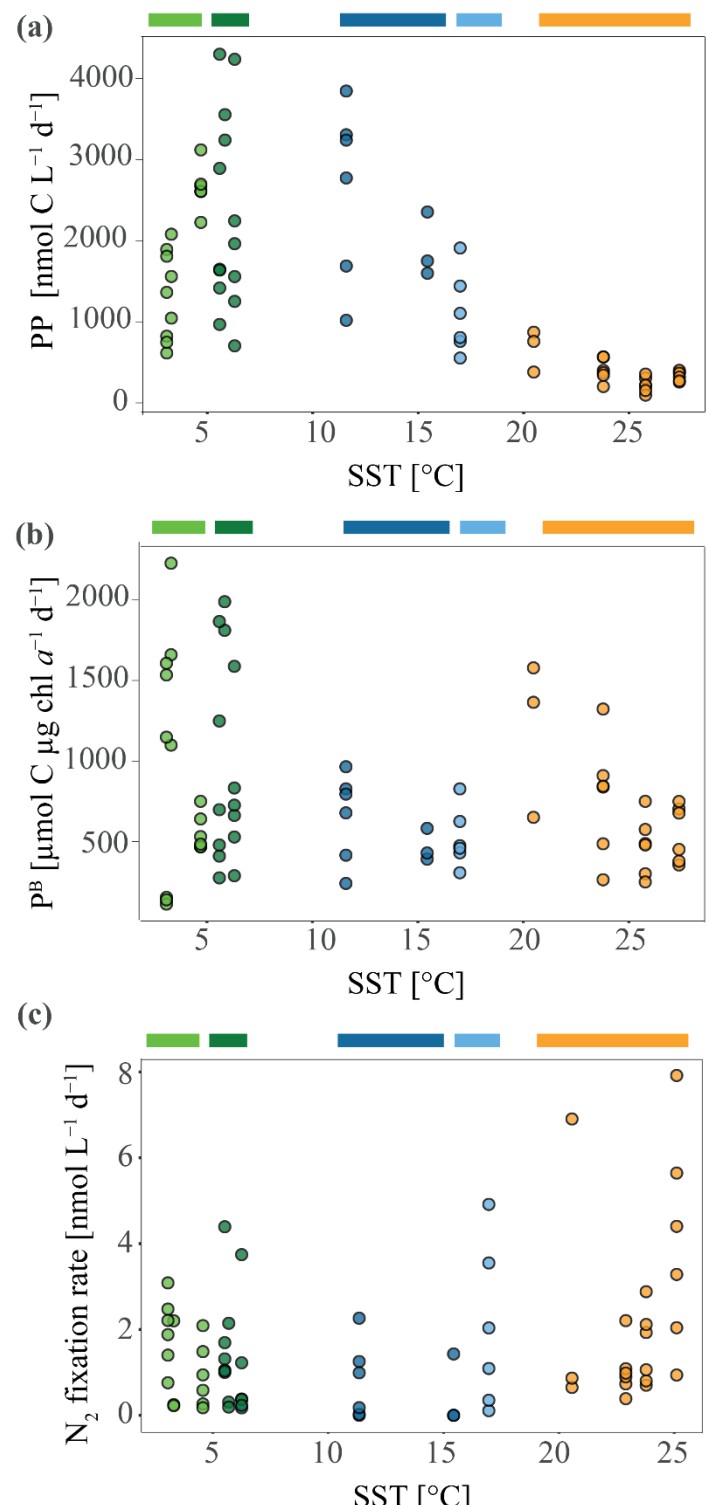




**Figure 3: Primary productivity (PP) and specific primary productivity (P$^B$) measured during the MD206 cruise. (a) PP in micromole**
**per liter and day against sea surface temperature (SST) in °C. (b) P$^B$, normalized by chl $a$ concentration. (c) Nitrogen fixation rates**
**against sea surface temperature (SST) in degree centigrade measured during the MD206 cruise. Rates are shown in nanomol per**
**liter and day; Colored bars indicate water masses; yellow bar highlight the Indian Ocean gyre (ISSG), light blue bar the Subtropical**
**Front (STF), blue bar the Subantarctic Front (SAF), dark green bar the Polar Front Zone (PFZ) and light green bar marks Antarctic**
**Zone (AZ).**




**Figure 4: (a) Eukaryotic and (b) prokaryotic community structures of different water masses measured during the MD206 cruise.**
**Redundancy analysis (RDA) of 18S and 16S rRNA gene ASV tables as response variables and environmental metadata as**
**explanatory variables; environmental metadata are represented as arrows. Constrained Analyses were performed by water mass.**
**There were significant relationships between water masses and community dissimilarities (PERMANOVA, 999 permutations; p <**
**0.001, R$^2$ = 0.67 for eukaryotes and p < 0.001, R2 = 0.74 for prokaryotes, respectively). Colors indicate major water masses according**
**to the legend; yellow bar highlights the Indian South Subtropical Gyre (ISSG), light blue bar the Subtropical Front (STF), blue bar**
**the Subantarctic Front (SAF), dark green bar the Polar Front Zone (PFZ) and light green marks the Antarctic Zone (AZ).**



Eukaryotic (c) and prokaryotic (d) general dissimilarity model (GDM) with (I) observed compositional dissimilarity against predicted ecological distance, calculated from temperature + dissolved oxygen + $NO_3^-$ + $NH_4^+$ + Si + chl $a$ + PP + $N_2$ fixation, (II) observed compositional dissimilarity against predicted compositional dissimilarity to test the model fit, and contribution of (III) PP and (IV) $N_2$ fixation to community dissimilarity expressed as a function of the environmental parameter (f(PP) and f(N2fix), respectively). For all functional plots of environmental data of the GDM analysis see Fig. S11. Eukaryotic (e) and prokaryotic (f) UpSet plots of ASV intersections between Longhurst provinces. Analyses are based on binary tables (presence/ absence) and the sum of all ASVs found across samples within one province. Intersection size shows number of ASVs shared between provinces (black dots, associated) and ASVs only found in one province (only black dot). Set size shows number of ASVs found in a specific Longhurst province.





**Table 1. Sampling stations visited during the MD206 cruise, including chlorophyll *a* concentrations, primary productivity (PP), specific primary productivity (P^B), and N₂ fixation. Mixed layer depth (MLD) was calculated using Δd = 0.03 kg m⁻³ compared to a surface reference depth of 7 m. NA indicates no data. Ranges and mean for sample replicates of N₂ fixation and PP are given (n = 3 for stations 3, 9, 11, 15; n = 6 for stations E, 37, 2, 4, 6, 7, 14, 16, 18). Colored bars indicate water masses; yellow bar highlight the Indian Ocean gyre (ISSG), light blue bar the Subtropical Front (STF), blue bar the Subantarctic Front (SAF), green bar the Polar Front Zone (PFZ), and dark green bar the Antarctic Zone (AZ)**

| Station | Longitude [°E] | Latitude [°S] | MLD [m] | chl *a* [µg L⁻¹] | Primary productivity (PP) [µmol C L⁻¹ d⁻¹] | specific PP (P^B) [µmol C µg chl *a*⁻¹ L⁻¹ d⁻¹] | N₂ fix [nmol L⁻¹ d⁻¹] | MQR [nmol L⁻¹ d⁻¹] |
|---|---|---|---|---|---|---|---|---|
| 37 | 52.003 | 55.004 | 52.5 | 4.96 | 587.42 - 1875.58; 1185.59 | 118 - 1628; 795 | 0.76 - 3.09; 1.97 | 1.2 |
| 11 | 63.006 | 56.499 | 49.5 | 0.92 | 1020.91 - 2065.12; 1541.95 | 1115 - 2255; 1683 | 0.23 - 2.20; 0.89 | 1.2 |
| 10 | 68.421 | 50.667 | 88.2 | NA | NA | NA | NA | NA |
| E | 72.367 | 48.8 | 81.3 | 4.09 | 2212.37 - 3114.53; 2645.72 | 477 - 762; 567 | 0.18 - 2.09; 0.92 | 0.7 |
| 7 | 58.004 | 47.667 | 49.6 | 3.33 | 942.99 - 4305.26; 2129.45 | 283 - 1889; 843 | 1.0 - 4.39; 1.75 | 1.2 |
| 9 | 64.999 | 48.501 | 69.4 | 1.76 | 3236.8 - 3553.33; 3395.07 | 1834 - 2013; 1924 | 0.19 - 2.15; 0.88 | 0.8 |
| 6 | 52.102 | 45.000 | 41.7 | 2.28 | 676.44 - 4242.33; 1977.6 | 296 - 1609; 784 | 0.17 - 3.25; 0.93 | 0.9 |
| 14 | 74.884 | 42.499 | 30.8 | 3.93 | 994.1 - 3847.07; 2635.94 | 248 - 979; 665 | 0.0 - 2.26; 0.78 | 0.7 |
| 15 | 76.407 | 39.999 | 29.8 | 3.95 | 1579.92 - 2341.93; 1884.88 | 400 - 593; 477 | 0.0 - 1.43; 0.24 | 1.2 |
| 4 | 52.79 | 40.001 | 54.6 | 2.23 | 524.32 - 1876.67; 1069.21 | 315 - 841; 531 | 0.11 - 4.91; 2.01 | 3.5 |
| 3 | 53.499 | 35.000 | 15.9 | 0.53 | 350.33 - 845.86; 642.59 | 662 - 1599; 1215 | 0.65 - 6.91; 2.81 | 5.4 |
| 16 | 73.466 | 35.001 | 19.9 | 0.40 | 170.05 - 537.91; 378.28 | 271 - 1341; 790 | 0.39 - 2.21; 1.05 | 1.3 |
| 2 | 54.1 | 30.001 | 12.9 | 0.55 | 63.24 - 324.72; 190.38 | 257 - 762; 484 | 0.7 - 2.88; 1.58 | 2.6 |
| 18 | 65.832 | 28.0 | 16.9 | 0.49 | 226.09 - 371.07; 301.3755 | 364 - 762; 563 | 0.94 - 7.92; 4.04 | 5.0 |