# Peer review of "Hydrographic fronts shape productivity, nitrogen fixation, and"

_Biogeosciences, 2021_

## Author Response (AR1)

Dear Koji Suzuki,

Thank you very much for your comment. We carefully revised our manuscript. Please find the detailed responses to the reviewers below including the line numbers (where appropriate) with the revised changes.

**Reviewer 1**
We thank the reviewer for their time, valuable comments and suggestions. We are pleased to receive positive feedback on our manuscript. We will integrate the comments made by the reviewer in our revised version of the manuscript. Please see below for our detailed response to all comments:

- L425: "negative correlation" Is it significant in the ISSG region? Maybe, Fig. S7 (e) shows the significance for all sampling data. Recheck it.

We thank the reviewer for their remark. Yes, the p-value was 0.02 in that case. We added "significant" negative correlation to the text. [L. 426]

- L439-443: It will be better to discuss about other bacterial lineages according to results of the SIMPER (e.g. Gammaproteobateria SUP05 (contributor for dissimilarity between SSTC-SO)).

We thank the reviewer and we will extend our discussion as follows [L. 440 - 451]:

Further, our results show that phytoplankton community structure appears to be tightly coupled to the occurrence of specific heterotrophic organisms (Table S6), and thus may mediate an indirect effect of PP through microbial food web (as also noted in, e.g., Sarmento and Gasol 2012).

For example, in areas of relatively high diatom concentrations, we found increased proportions of Flavobacteria. These bacteria specialize on successive decomposition of algal-derived organic matter (Teeling et al., 2012) and are known associates of diatoms (Pinhassi et al., 2004). Further, Planktomarina belonging to the Roseobacter RCA subgroup had relatively high proportions in the SO and is generally suggested to occur in colder environments (Giebel et al. 2009) and previously detected in the Polar Front (Wilkins et al. 2013). The RCA subgroup is known for DMSP degradation in phytoplankton blooms (Han et al. 2020).

In addition to bacteria known to be associated with phytoplankton, we also observed those which symbiose with other organisms (e.g. Georgieva et al. 2020), such as the sulphur oxidising Thioglobaceae (SUP-05 cluster), previously found in symbiosis with Myctophidae

fish near Kerguelen (Gallet et al. 2019). While beyond the scope of this study, we encourage further investigations of such trans-kingdom functional interactions as they themselves may offer regional insights.

Additional References:

1. Giebel et al. 2009; https://doi.org/10.1111/j.1462-2920.2009.01942.x

2. Wilkins et al. 2013; https://doi.org/10.1111/1574-6976.12007

3. Georgieva et al. 2020 https://doi.org/10.3389/fmicb.2020.01636

4. Gallet et al. 2019 https://doi.org/10.1371/journal.pone.0226159

- L459: "So" -->"SO" ?

We thank the reviewer for detecting this mistake and note we have changed as suggested. [L. 467]

Figures

- Fig. 4 c and d: Add the units of x axis.

We thank the reviewer for their comment. We will add the units in the revised version of our manuscript.

- L292, L330-331: Revise the unit (umol C L-1d-1) according to those in Fig. 4c and d. It is confusing.

We thank the reviewer for their comment. We note that we performed the gdm analysis both on the original data and z-scored metadata and thus had different scales in our manuscript and the figure. We will correct the units accordingly.

Tables

- L806-808: Explanation about colored bars is not needed. Remove it.

We thank the reviewer for their comment and note that we have removed this information in the revised version of our manuscript,

Supplementary information

- Mismatch of the supplementary figures and figure legends. Where is the legend of Fig. S4 about PN : Chl. a ratio?

We thank the reviewer for catching this mismatch and note that we will correct and add the figure legend in the revised version of the supplement.

- Fig. S4: Color bars above the figure are not linked legend symbol's colors. It's confusing. Revise them.

We thank the reviewer for detecting this mistake, we will correct the colors accordingly.

- Fig. S7: Color bars (indicate ISSG, STF, SAF…etc) above figure are valuable for figures (a, b, c, d), but not for figures (e, f, g, h). I think it's better to add the legend symbols to fig (e~g).

We thank the reviewer for their advice and will add the legend symbols in the revised figure.

**Reviewer 2**

We thank the reviewer for their time and positive comments on our manuscript. Please find below our detailed responses.

- Figure 4: Is the UpSet plot analysis described in the methods?

Yes, we used the upsetR package in R and described the analysis in our method section (Line 205-207).

- Line 378: …, "and" biological regional N2 fixation might become an important N-source for productivity

We thank the reviewer for this comment and will correct this in the revised version of our manuscript. [L. 379]

- I'm surprised that you didn't find any Synechococcus in the Southern Ocean…did you do any flow cytometry measurements?

We thank the reviewer for this comment. We note that we were not able to detect any Synechococcus ASVs in the SO samples. However, we did find low concentrations of Zeaxanthin in our HPLC analysis, whis is an indicator for prokaryotes (Fig. S5). Unfortunately, our Flow Cytometry samples were compromised during our sample processing and are thus not able to include them in this dataset.